# Age and Post-Lobectomy Recurrence after Endoscopic or Robotic Thyroid Surgery: A Retrospective Cohort Study of 2348 Papillary Thyroid Carcinoma Patients

**DOI:** 10.3390/cancers15235506

**Published:** 2023-11-21

**Authors:** Jin-Seong Cho, Yong-Min Na, Hee Kyung Kim

**Affiliations:** 1Department of Surgery, Chonnam National University Medical School, 42 Jebong-ro, Dong-gu, Gwangju 61469, Republic of Korea; 2Department of Internal Medicine, Chonnam National University Medical School, 42 Jebong-ro, Dong-gu, Gwangju 61469, Republic of Korea; albeppy@jnu.ac.kr

**Keywords:** age distribution, thyroid neoplasms, thyroidectomy, recurrence, risk factors

## Abstract

**Simple Summary:**

The biology of papillary thyroid carcinoma (PTC) in young patients is poorly understood, and there are conflicting data regarding recurrence for younger patients compared to older patients. We retrospectively analyzed 2348 clinically node-negative (cN0) PTC patients who underwent a thyroid lobectomy between 2008 and 2017. The clinicopathological characteristics and oncologic outcomes of young patients were compared to older patients. In the young age group, there was a significantly larger proportion of females, endoscopic/robotic thyroid lobectomy, large tumor sizes, and stage N1a. Post-lobectomy recurrences were higher in the young age group. In the Cox analysis, young age, large tumors, and stage N1a were significant risk factors. The multivariate analyses reveals that young age and stage N1a are significant risk factors. Conversely, minimally invasive and endoscopic/robotic thyroidectomies were not risk factors for post-lobectomy recurrence. Further studies are needed to elucidate the relationship between young age and the risk of post-lobectomy recurrence.

**Abstract:**

The biology of papillary thyroid carcinoma (PTC) in young patients is poorly understood, and there are conflicting data regarding the recurrence for younger patients compared to older patients. We retrospectively analyzed 2348 clinically node-negative (cN0) PTC patients who underwent a thyroid lobectomy between 2008 and 2017. Young age was defined as less than 35 years old. The clinicopathological characteristics and oncologic outcomes of the young age group were compared to those of the older age group. The number of young age cN0 PTC patients accounted for 20.7% of the enrolled patients, and 24.2% were upstaged into pathologic N1a. The young age group had a significantly larger proportion of females, endoscopic/robotic thyroid lobectomies, stage N1a, and larger tumor sizes. Post-lobectomy recurrences were significantly higher in the young age group. In the Cox analysis, young age, large tumor size, and stage N1a were significant risk factors. The multivariate analysis reveals that young age and stage N1a are significant risk factors. Conversely, minimally invasive or robot-endoscopic thyroidectomies were not risk factors for post-lobectomy recurrence compared to conventional thyroidectomies. While young patients with a stage N1a had a significant risk factor for post-lobectomy recurrence, endoscopic/robotic thyroidectomy was as feasible and safe as conventional thyroidectomies in the median seven-year oncologic follow-up. Further high-quality studies are needed to elucidate the relationship between age and the risk of post-lobectomy recurrence.

## 1. Introduction

Thyroid cancer was ranked ninth in global cancer incidence in 2020 and has since steeply risen over the last decades in the United States of America [1,2]. The incidence in Korea also has increased, especially in males and young individuals [3]. Young patients with papillary thyroid cancer (PTC) are perceived to have a low-risk disease and a favorable prognosis, even when faced with advanced locoregional disease [4,5].

There has been a major change in the Eighth American Joint Committee on Cancer (AJCC) staging system, where 55 years old now represents the new age cutoff point [6]. In this system, patients younger than 55 with PTC are considered to have only two possible stages based on the absence (stage I) or presence (stage II) of distant metastases; thus, PTC patients with lymph node metastases are considered to have stage I PTC [7]. However, it remains controversial whether relapse in younger PTC patients with lymph node metastasis has a favorable prognosis, unlike the older age group.

The impact of young age on the recurrence and prognosis of PTC is poorly understood, and there are conflicting data compared to older patients. Adam et al. demonstrated that the presence and number of metastatic nodes in young PTC patients are associated with compromised survival [8]. However, Zaydfudim et al. reported that lymph node involvement did not influence the overall survival for patients younger than 45, whereas, in patients 45 years or older, lymph node involvement was associated with decreased overall survival [4].

Regional recurrence frequently occurs in young PTC patients, although the mortality rates are low [9]. We previously showed that patients younger than 35 had better prognoses despite frequent recurrences; however, patients older than 62.5 had worse prognoses [5]. We also showed that thyroid lobectomy was not a risk factor for recurrence compared to total thyroidectomy; however, we did not evaluate the impact of various surgical approaches as risk factors for recurrence.

Robotic or endoscopic thyroidectomy (RET) has gained popularity for the treatment of low- to intermediate-risk cN0 PTC. RET offers a distinct advantage over conventional or minimally invasive surgeries of the absence of a visible neck scar [10]. The desire for and proportion of RETs are increasing in young patients since a major concern of thyroid surgeries, especially in younger females, is neck scarring. However, the most important issue in the treatment of thyroid cancer is the oncologic outcome, and it should not be neglected or overlooked in favor of cosmetic or functional outcomes. However, studies on the oncologic outcomes of RET, such as locoregional recurrence and prognosis, are very limited.

Identifying the oncologic indicators for age, which are biologically different, is very important in deciding an appropriate therapeutic strategy [5]. This study aimed to identify the impact of age on post-lobectomy recurrence and to compare the oncologic outcomes of minimally invasive thyroidectomy and RET compared to conventional thyroidectomy.

## 2. Materials and Methods:

### 2.1. Patients and Inclusion/Exclusion Criteria

A retrospective review was performed on the medical records of 10,402 consecutive thyroid surgeries by five endocrine surgeons in Gwangju and Hwasun Chonnam National University Hospital from 2008 to 2017. A total of 2873 (27.6%) patients who had undergone a thyroid lobectomy were recruited. Finally, a total of 2348 clinically node-negative (cN0) low- to intermediate-risk patients were enrolled and analyzed after excluding 525 patients (Figure 1). Exclusion criteria included patients less than 18 years old, the presence of non-papillary thyroid cancers, and insufficient clinical data. Two patients with distant metastases confirmed after the immediate completion with total thyroidectomy were excluded. Patients who underwent immediate completion within 12 months were defined as having a persistent disease and were excluded. In addition, 96 cases that had a loss of follow-up within the first 12 months were excluded. The clinicopathological characteristics and oncologic outcomes included patient age at diagnosis, sex, surgical approach, pathologic characteristics with tumor size, extrathyroidal extension (ETE), and lymph node metastasis. In cN0 PTC patients with post-lobectomy recurrences, the postoperative dynamic risk stratification (DRS) and supplementation with levothyroxine (LT4) were also analyzed.

### 2.2. Surgical Approach

Thyroid lobectomies were conducted via three methods in our two institutions: conventional, minimally invasive, and RET. Minimally invasive thyroid lobectomy was defined as an incision of 3.0 cm or less. RETs using only the bilateral axillary and breast (BABA) approach were included. Our institutions started using the BABA endoscopic approach in August 2006 and introduced the TAA (Transaxillary approach) in February 2008. Only 13 cases of TAA were conducted since we preferred the BABA approach. We presume that BABA is superior for oncologic outcomes, especially in total thyroidectomies. Since May 2008, only the BABA endoscopic approach has been primarily implemented, and the BABA approach by a robot using the DaVinci Xi system has been conducted since 2010. Transoral endoscopic (2020) and Transoral robotic thyroidectomy (2021) were conducted after the periods of this study (2008–2017). Trans-axillary and trans-oral RETs were excluded because the number of cases was insufficient, and neither approach had sufficient follow-up periods.

### 2.3. Central Neck Dissection (CND)

Prophylactic ipsilateral central neck dissection (CND) and intraoperative frozen biopsies have been used previously with good results [11]. Raffaelli et al. reported that prophylactic ipsilateral CND and frozen section biopsies in cN0 patients can change the extent of the thyroidectomy in about one-fourth of the patients scheduled for a thyroid lobectomy [12]. If the intraoperative stage N was upstaged to cN1a, even if the patient was preoperatively cN0, prophylactic ipsilateral CND and frozen biopsies were performed. Frozen section examinations are a safe and effective technique to decrease the need for an immediate-completion thyroidectomy and recurrence. If the frozen section was positive for macro-metastases, we converted to a total thyroidectomy. Additionally, if the frozen section had micro-metastasis, we used a wider compartmental CND or immediately converted to a total thyroidectomy.

### 2.4. Risk Stratification

Low-risk was defined as patients with T1, T2, pN0-staged classic, or less aggressive variant PTC, and intermediate-risk was defined as patients with any T3, N1a, aggressive variant, or lymphovascular invasion. Pathologic stage N1a patients with central metastasis made up 16.2% of all the patients, even though thyroid lobectomies were performed only in the preoperatively screened cN0 patients.

### 2.5. Immediate vs. Delayed Completion

In total, within 12 months, 24 immediate-completion thyroidectomies were conducted, and 11 patients had remaining cancers on the contralateral thyroid or central compartment, while 13 patients had no remaining cancer lesions. About 45.8% of the patients with an immediate-completion thyroidectomy had residual thyroid or lymph node metastases, and this proportion was comparable to a previous study [13].

### 2.6. Follow-Up

Every visit, thyroid hormone, thyroglobulin, and anti-thyroglobulin antibody levels were measured, and at six-month or one-year intervals, a neck ultrasonography was performed. The dose of LT4 supplementation was regulated to maintain the optimal thyroid-stimulating hormone (TSH) level and was tapered or discontinued by DRS. Tapering or cessation of LT4 was attempted if the optimal TSH level was maintained while the risk of post-lobectomy recurrence decreased after DRS [14,15]. Recurrence was confirmed by aspiration cytology with wash-out thyroglobulin and postoperative histology. If the recurrence was confirmed within 12 months in patients who underwent an immediate-completion thyroid lobectomy, the residual cancer was presumed to be caused by an inappropriate thyroid lobectomy and defined as a persistent disease rather than a recurrence.

### 2.7. Statistics

Using the area under the receiver operating characteristic curve (AUC-ROC), the cutoff values of the continuous variables were identified. Cox proportional regression models were used to calculate the hazard ratios (HRs) and to assess the simultaneous effects of various predictive factors associated with post-lobectomy recurrence. The statistical difference in recurrence rate was compared using a log-rank test and Kaplan–Meier analysis. Statistical significance was indicated by a two-sided *p*-value less than 0.05. In addition, the factors with an HR over 1.0 were considered statistically significant. Analyses were performed using IBM Statistical Product and Service Solutions Statistics version 27.0 (IBM Corp., Armonk, NY, USA).

### 2.8. Ethical Approval

This study was approved by the Institutional Review Board of Chonnam National University Hospital (CNUH-2023-029) and Chonnam National University Hwasun Hospital (CNUHH-2023-051), which waived the need for informed consent.

## 3. Results

### 3.1. Baseline Characteristics of cN0 PTC Patients

A total of 2348 thyroid lobectomy patients with stage cN0 were enrolled in this study (Table 1). The median age was 44 and ranged between 18 and 77 years old. Minimally invasive thyroidectomies (33.7%) and RETs (20.0%) were compared to conventional thyroidectomies (46.3%). The proportion of T3 was 10.6% (*n* = 248), and pathologic N1a was 16.2% (*n* = 380). Other T or N stages were as follows: T1a (82.3%), T1b (6.1%), T2 (1.0%), N0 (44.6%), and Nx (39.2%). According to histology and DRS, 1789 (76.2%) low-risk and 559 (23.8%) intermediate-risk patients were identified. The proportion of patients with ongoing LT4 supplementation at the last follow-up was 53.8%.

During the follow-up period (mean: 83.9 months), there were sixty-two (2.6%) recurrences, and the observed recurrence rates per patient’s age category are depicted in Figure 2A. The age cutoff value for post-lobectomy recurrence was 35 years (AUC-ROC = 0.673) (Figure 2B), and the data were further analyzed after dividing the patients into two groups: the young age group and the older age group.

### 3.2. Characteristics of the Young Age Group

Table 2 shows the differences between the young age group (*n* = 487; 20.7%) and the older age group (*n* = 1861; 79.3%). The proportions of females (83.8% vs. 75.6%, *p* = 0.001) and RETs (43.5% vs. 13.8%, *p* = 0.001) were significantly larger in the young age group. The mean tumor size (0.82 cm vs. 0.74 cm, *p* = 0.001), the proportion of tumor size greater than 7.5 mm (57.3% vs. 51.1%, *p* = 0.015), and stage N1a (24.2% vs. 14.1%, *p* = 0.001) were significantly larger in the young age group. The proportion of ongoing LT4 supplementation at the last follow-up was not different between the two groups (52.6% vs. 54.2%, *p* = 0.529). During the follow-up period (mean: 83.9-month), the proportion of recurrences was significantly larger in the young age group than in the older age group (6.2% vs. 1.7%, *p* = 0.001).

### 3.3. Cox Analysis

A Cox regression was performed to identify the risk factors for post-lobectomy recurrence; the analysis results arranged with hazard ratios (HRs) are shown in Table 3. Young age (HR = 3.69), tumor size larger than 7.5 mm (HR = 1.83), and stage N1a (HR = 3.38) are significant risk factors for recurrence in the univariate analysis. Conversely, sex (*p* = 0.471), ETE (*p* = 0.150), and multifocality (*p* = 0.703) are not significant risk factors for post-lobectomy recurrence.

The multivariate analysis shows that young age (HR = 3.27; *p* = 0.001) and stage N1a (HR = 2.04; *p* = 0.014) are statistically significant risk factors for post-lobectomy recurrence. However, tumor sizes larger than 7.5 mm (HR = 1.37; *p* = 0.257) and surgical approaches, such as minimally invasive thyroidectomy (HR = 1.45, *p* = 0.233) or RET (HR = 1.31; *p* = 0.464), are not statistically significant risk factors for post-lobectomy recurrence.

### 3.4. Kaplan–Meier Analysis

The above factors were further analyzed using the Kaplan–Meier method (Figure 3). Young age (<35 years) (log-rank *p* = 0.001) and stage N1a (log-rank *p* = 0.001) are potent risk factors for post-lobectomy recurrence (Figure 3A,B). Conversely, minimally invasive thyroidectomy and RET, rather than the conventional approach (Log-rank *p* = 0.241), are not risk factors for post-lobectomy recurrence (Figure 3C).

## 4. Discussion

Thyroid cancer is the only adult cancer that uses age as a prognostic factor in the TNM staging system. The eighth AJCC staging system concluded that being 55 years or older was a significant risk factor for prognosis. Incorporating this new cutoff age revealed that the HR of cancer-related deaths between the younger and older age groups decreased to about 4, compared with an HR of 8 using the former cutoff age of 45 years [7]. However, most staging systems that use age to stratify risk tend to be inaccurate in predicting recurrence-free survival, since younger patients have higher recurrence rates. Additionally, these staging systems have been derived from multivariate analyses that do not consider recurrence or the effect of therapy [16].

Our results show that there are more recurrences in PTC patients younger than 35 years old than in older patients. Young age remained a significant predictor of recurrence even after controlling for differences in the distribution of potential prognostic factors. The HR of post-lobectomy recurrence for patients younger than 35 years old was 3.69, and there was no cancer-related death. This study focused solely on recurrence in consideration of age, especially in low- to intermediate-risk cN0 PTC patients who had undergone a thyroid lobectomy. The present findings support previous reports showing that thyroid cancer at a younger age has a favorable prognosis despite frequent recurrences compared with older counterparts.

Recently, endocrine surgeons have been performing more thyroid lobectomies on low-risk differentiated thyroid cancers and less common completion thyroidectomies since the introduction of the 2015 ATA guidelines [17,18,19]. Disease-free status and survival cannot be assured at the low stage in most systems, thus, providing imperfect guidance in selecting the therapy [16,20]. Furthermore, the literature on oncologic outcomes, such as recurrence or prognosis, after RET is very limited. 

In other studies, oncologic outcomes, including disease-specific survival and recurrence rates, were not significantly different between RETs and conventional thyroidectomies, although the follow-up period was rather short [21,22]. However, those reports suffered from limitations, including a small sample size, a short study period, and a lack of information about the definite HR per different surgical options on recurrence. Furthermore, these studies included a heterogeneous case population of all surgery types ranging from lobectomy to total thyroidectomy and even included patients with lateral neck dissection with radioactive iodine therapy.

This study is the latest to compare the recurrence of thyroid cancer in young patients with a homogenous case population that included only cN0 patients who had undergone a thyroid lobectomy. Additionally, all data used in this study were from patients of the same ethnicity who underwent the same surgical and follow-up strategies at two institutions over a very recent period (median of seven to fourteen years). In addition, this study included a multivariate analysis of the differences of three surgical approaches: conventional, minimally invasive, and RET thyroidectomies.

Currently, the reason large tumor size is a risk factor for recurrence is unclear, with some evidence suggesting it is significant [23,24] and other evidence suggesting the contrary [25]. As for a tumor size of 1–4 cm, its prognostic importance is also controversial. Many studies have reported positive results; however, others have reported negative results [23,25]. This study found a significant difference in the young age group regarding female predominance, large tumor size, and increased lymph node metastasis. Young age and lymph node metastasis were indicated by the multivariate analysis to be independent risk factors for post-lobectomy recurrences. However, tumors larger than 7.5 mm were found not to be a risk factor. The importance of large tumor size is inconclusive and should be further investigated in subsequent studies. Using this dataset, the prognostic significance of tumor size will be investigated further via a follow-up independent analysis.

We found that ETE, multifocality, and LT4 supplementation were not significant independent predictors of post-lobectomy recurrence, although these were not major focuses of this study. Contrary to other studies [26,27,28,29], multifocality was not a risk factor for post-lobectomy recurrence. Multifocality should not be considered an indication for immediate completion in patients with precise preoperative screening and selected DTC who did not have cancer foci in the opposite lobe. We presume our results are plausible given the advanced skill and effort to find bilateral cancer in the remnant thyroid lobe and due to high-resolution ultrasonography.

Even after controlling for differences in the distribution of potential prognostic factors, young age with stage N1a was an independent predictor of post-lobectomy recurrence. The underlying biology of thyroid cancer in young patients need to be elucidated, and the development of tailored treatments for this patient population is crucial. Our findings suggest that young patients present with more aggressive features and higher recurrence rates compared to older patients and should be carefully treated from initial evaluation to surgery and through postoperative care.

This retrospective study has several inherent limitations. First, the sample population may not represent larger populations, resulting in limited generalizability. The second is institutional bias, reflecting patients with more favorable outcomes according to treatment bias, even though the sample size of this study was moderate in our two tertiary hospital-based cohorts. Our previous study could not determine the definite cutoff age for locoregional recurrence (AUC = 0.486) even though the cutoff age for PTC-related deaths was 62.5 years old [5]. However, we acknowledge that our previous study was more heterogeneous and was composed of various risk patients who had undergone various surgical treatment options, such as thyroid lobectomy, total thyroidectomy, and lateral neck dissection with radioactive iodine therapy. This study also focused solely on post-lobectomy recurrence in low- to intermediate-risk cN0 PTC patients and identified the relationship between young age and post-lobectomy recurrence.

The third limitation is selection bias. In our cohort, the proportion of T3 was 10.6% (*n* = 248), and N1a was 16.2% (*n* = 380), even though we enrolled previously screened low- to intermediate-risk patients. Recently, endocrine surgeons have performed more thyroid lobectomies for low- to intermediate-risk thyroid cancer and less common completion thyroidectomy since the introduction of the 2015 ATA guidelines [17,18]. Future multi-institutional, large cohort, and long-term studies are needed.

The fourth is the crossing survival curves that occurred in the Kaplan–Meier analysis around 120 months or later (Figure 3C). We presumed that this phenomenon was derived from incomplete surgical skills in the early learning periods of RET. In the beginning and implementation periods of RET in our institutions between 2008 and 2010, incomplete CND was more prevalent among the less-trained or less-experienced surgeons. The crossing survival curves could be the point of the development of acceptable surgical skills, and completion of oncologic safety. Crossing survival curves indicate that the recurrence-free probabilities of RET compared to minimally invasive or conventional thyroidectomies are equal at one or more time points [30]. However, whether recurrence continues to be maintained should be tracked for a longer period.

## 5. Conclusions

While young age (<35 years old) with stage N1a was a significant risk factor for post-lobectomy recurrence, minimally invasive and RET thyroidectomies were as feasible and safe in the oncologic outcomes (median 7 years) as conventional thyroidectomies. Further studies with a long-term follow-up and large patient sample size are needed to assess the ultimate long-term oncologic outcomes of RET in young patients.

## Figures and Tables

**Figure 1 cancers-15-05506-f001:**
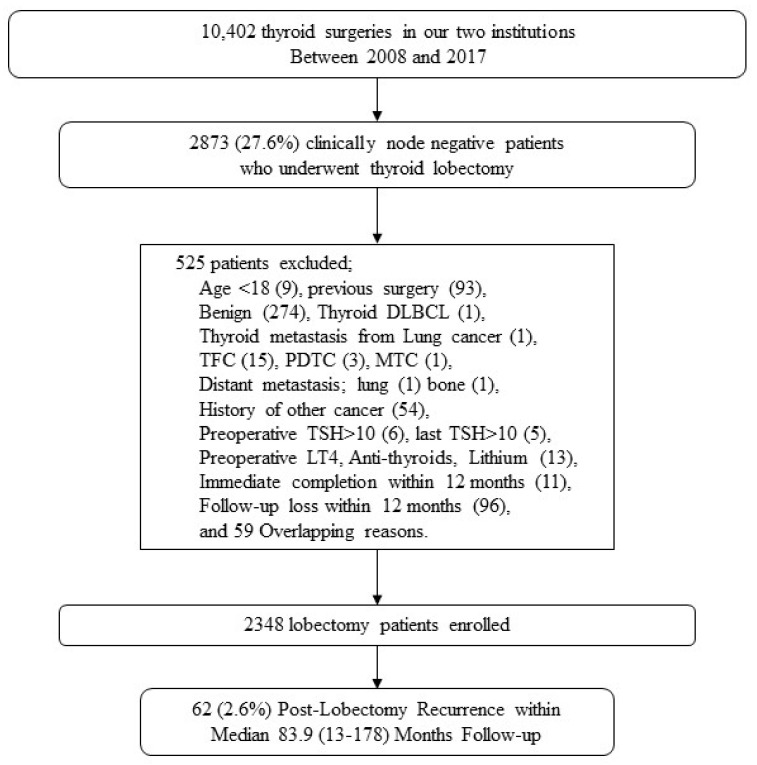
Flow chart of the stepwise selection of the study population. DLBCL, diffuse large B-cell lymphoma; TFC, thyroid follicular carcinoma; PDTC, poorly differentiated thyroid carcinoma; MTC, medullary thyroid cancer; TSH, thyroid-stimulating hormone; LT4, levothyroxine.

**Figure 2 cancers-15-05506-f002:**
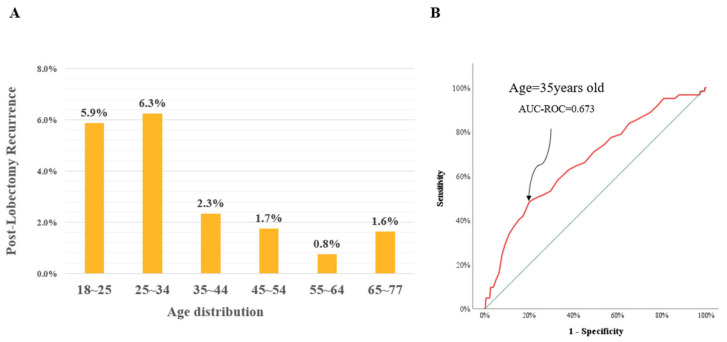
Post-lobectomy recurrence by age group. (**A**) Observed post-lobectomy recurrence per patient’s age for the papillary thyroid cancer patients. (**B**) The age of 35 years was defined as the optimal cutoff value in the ROC curve analysis with the highest Youden’s index.

**Figure 3 cancers-15-05506-f003:**
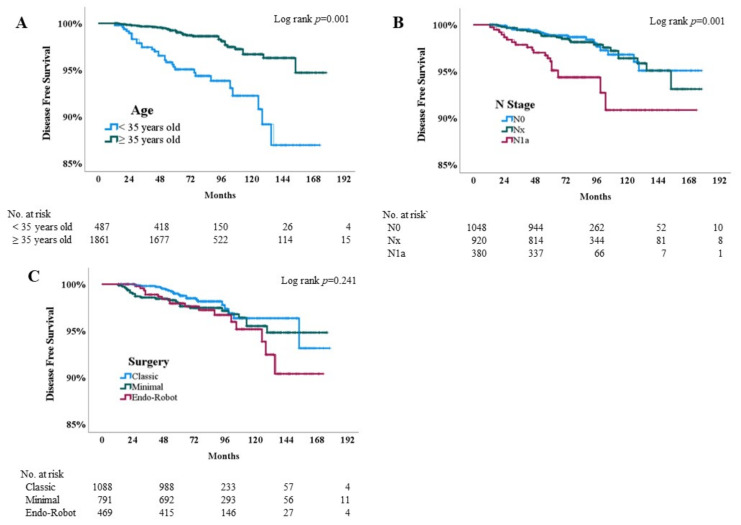
Kaplan–Meier curves for post-lobectomy recurrence stratified by various risk factors. (**A**) Young age (<35 years) (log-rank *p* = 0.001) and (**B**) stage N1a (log-rank *p* = 0.001) were potent risk factors for post-lobectomy recurrence. (**C**) Minimally invasive and RET thyroidectomies were not risk factors compared with conventional thyroidectomies (log-rank *p* = 0.241).

**Table 1 cancers-15-05506-t001:** Baseline characteristics of cN0 PTC patients who underwent thyroid lobectomy.

Characteristics	Total n = 2348
**Age**	
Median (range), years	44 (18–77)
**Sex**	
Male	534 (22.7%)
**Surgical approach**	
Conventional	1088 (46.3%)
Minimally invasive	791 (33.7%)
RET	469 (20.0%)
**T stage**	
T1a	1932 (82.3%)
T1b	144 (6.1%)
T2	24 (1.0%)
T3	248 (10.6%)
**CND**	1428 (60.8%)
**N stage**	
N0	1048 (44.6%)
Nx	920 (39.2%)
N1a	380 (16.2%)
**ATA risk**	
Low-risk	1789 (76.2%)
Intermediate-risk	559 (23.8%)
**Follow-up**	
Mean ± SD, months	83.9 ± 33.4
Median (range)	80 (13–178)
**LT4**	
No supplementation	1084 (46.2%)
Supplementation	1264 (53.8%)
**Recurrence**	
Recurrence	62 (2.6%)

RET, robotic or endoscopic thyroidectomy; CND, central neck dissection; ATA, American Thyroid Association; LT4, levothyroxine; SD, standard deviation.

**Table 2 cancers-15-05506-t002:** Characteristics of young (<35 years old) patients with papillary thyroid cancer.

Characteristics	Age < 35 n = 487 (20.7%)	Age ≥ 35 n = 1861 (79.3%)	Total n = 2348	*p*-Value
**Age**				
Median (range), years	31 (18–35)	46 (36–77)	44 (18–77)	0.001
**Sex**				
Female	408 (83.8%)	1406 (75.6%)	1814 (77.3%)	0.001
Male	79 (16.2%)	455 (24.4%)	534 (22.7%)	
**Surgical approach**				
Conventional	141 (29.0%)	947 (50.9%)	1088 (46.3%)	0.001
Minimally invasive	134 (27.5%)	657 (35.3%)	791 (33.7%)	
RET	212 (43.5%)	257 (13.8%)	469 (20.0%)	
**Tumor size**				
Mean ± SD, cm	0.82 ± 0.50	0.74 ± 0.40	0.75 ± 0.42	0.002
Median (range), cm	0.7 (0.1–4.6)	0.7 (0.1–6.9)	0.7 (0.1–6.9)	
**Tumor Size**				
<7.5 mm	208 (42.7%)	910 (48.9%)	1118 (47.6%)	0.015
≥7.5 mm	279 (57.3%)	951 (51.1%)	1230 (52.4%)	
**ETE**	38 (7.8%)	210 (11.3%)	248 (10.6%)	0.026
**Multifocality**	39 (8.0%)	257 (13.8%)	296 (12.6%)	0.001
**CND**	310 (63.7%)	1118 (60.1%)	1428 (60.8%)	0.150
**N stage**				
N0	192 (39.4%)	856 (46.0%)	1048 (44.6%)	0.001
Nx	177 (36.3%)	743 (39.9%)	920 (39.2%)	
N1a	118 (24.2%)	262 (14.1%)	380 (16.2%)	
**LT4 Supplementation**				
LT4 off	231 (47.4%)	853 (45.8%)	1084 (46.2%)	0.529
LT4 on	256 (52.6%)	1008 (54.2%)	1264 (53.8%)	
**Follow-up**				
Mean ± SD, months	83.8 ± 33.5	84.0 ± 33.4	83.9 ± 33.4	0.907
Median (range)	80 (13–173)	79 (13–178)	80 (13–178)	
**Recurrence**				
Recurrence	30 (6.2%)	32 (1.7%)	62 (2.6%)	0.001

RET, robotic or endoscopic thyroidectomy; ETE, extrathyroidal extension; CND, central neck dissection; LT4, levothyroxine; SD, standard deviation.

**Table 3 cancers-15-05506-t003:** Cox regression analysis on covariates of post-lobectomy recurrence.

Covariate	n = 2348	Univariate		Multivariate	
		HR (95% CI)	*p*-Value	HR (95% CI)	*p*-Value
**Age**					
<35 years old	487	3.69 (2.24–6.08)	0.001	3.27 (1.89–5.64)	0.001
≥35 years old	1861	reference		reference	
**Sex**					
Female	1814	reference		reference	
Male	534	1.01 (0.55–1.86)	0.978	1.09 (0.75–2.07)	0.792
**Tumor status**					
Larger than 7.5 mm	1230	1.83 (1.09–3.08)	0.023	1.37 (0.79–2.37)	0.257
ETE	248	1.69 (0.83–3.43)	0.150	1.49 (0.71–3.10)	0.291
Multifocality	296	1.16 (0.55–2.43)	0.703	1.28 (0.60–2.73)	0.526
Thyroiditis	486	0.62 (0.31–1.25)	0.182	0.62 (0.30–1.29)	0.202
**Nodal status**					
N0	1048	reference		reference	
Nx	920	1.12 (0.60–2.07)	0.728	1.11 (0.59–2.07)	0.752
N1a	380	3.38 (1.81–6.29)	0.001	2.45 (1.27–4.73)	0.008
**LT4 Supplementation**					
LT4 off	1084	reference		reference	
LT4 on	1264	1.16 (0.69–1.94)	0.576	1.07 (0.62–1.86)	0.798
**Operation types**					
Conventional	1088	reference		reference	
Minimally invasive	791	1.35 (0.75–2.43)	0.320	1.45 (0.79–2.66)	0.233
RET	469	1.73 (0.91–3.28)	0.095	1.31 (0.64–2.66)	0.464

Cox regression with multivariate-adjusted HR, hazard ratio; CI, confidence interval; ETE, extrathyroidal extension; LT4, levothyroxine; RET, robotic or endoscopic thyroidectomy.

## Data Availability

All datasets generated and analyzed in this study are not publicly available but are available from the corresponding author on reasonable request.

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
