# Peer review of "Age and Post-Lobectomy Recurrence after Endoscopic or Robotic Thyroid Surgery: A Retrospective Cohort Study of 2348 Papillary Thyroid Carcinoma Patients"

_cancers, 2023, doi:10.3390/cancers15235506_

Round 1

Reviewer 1 Report (Previous Reviewer 1)

Comments and Suggestions for Authors

The manuscript is now sufficiently improved for publication

Author Response

All authors revised it and reviewed it by a specialized institution. And we quoted with 30 adequate references.

All of the authors thank for making it a better paper to all reviewers and editors.

Reviewer 2 Report (Previous Reviewer 2)

Comments and Suggestions for Authors

Thank you for allowing me to review this revised version of your manuscript. I don't have any major concerns with it.

On page 5 line 1, are you sure you don't mean "Institutional Review Board"?

Author Response

We modified the part you pointed out, and all authors revised it and reviewed it by a specialized institution. And we quoted with 30 adequate references.

All of the authors thank for making it a better paper to all reviewers and editors.

This manuscript is a resubmission of an earlier submission. The following is a list of the peer review reports and author responses from that submission.

Round 1

Reviewer 1 Report

Comments and Suggestions for Authors

This paper describes differences in clinical outcome between patients younger than 35 vs patients older than 35 years old. Although the authors used appropriate statistical methods to reach their results, regrettably I see no clinical relevance for their conclusions. Most thyroid cancer patients are routinely and successfully treated with total thyroidectomy with favourable prognosis and high survival rates. Also, the main conclusion of the paper, that “thyroid cancer at a younger age shows a favorable prognosis despite frequent recurrence compared with their older counterparts” sounds counterintuitive, while the authors offer no biological explanation for this phenomenon. Hence, I see no potential for these findings to be translated to clinical practice or to be of much use for the rest of the scientific community.

Comments on the Quality of English Language

Minor English editing required

Reviewer 2 Report

Comments and Suggestions for Authors

Thank you for the opportunity to review this paper. This is a very interesting retrospective review of different techniques for thyroid lobectomy and the different risks of recurrence. It is an interesting addition to the current literature where it was traditionally thought that thyroid lobectomy was not an oncological radical option for thyroid cancer.

Reviewer 3 Report

Comments and Suggestions for Authors

you have  a   good sample  , but some points  must be clarified 

"using only the bilateral axillary and breast (BABA) approach was included. Trans-axillary and trans-oral RET were excluded because the number of cases was insufficient, and both approaches did not have a sufficient follow-up period"  

If you chose just lobectomies why did you perform  bilateral axillary aparoach ?

Did you include or not  trans oral  approaches?

"We hold a positive attitude toward prophylactic ipsilateral central neck dissection and intraoperative frozen biopsy. Raffaelli et al. reported that prophylactic ipsi[1]lateral CND and frozen section biopsy in cN0 patients could change the extent of thy[1]roidectomy in about one-fourth of patients scheduled for thyroid lobectomy ..."

You explain a lot during the methods , it is better to explain and disccuss during discussion chapter, This way ,  the reading   becomes very hard . And you also performed neck dissections  with a lobectomy is controversial and must be discussed .

You did not explain why you  have excluded patients with tsh alterations

 at last : why dont  you perform a single question : Is The age important in recurrence after lobectomy ?

Of course your paper must  be published , but I guess you have to clarify the points above and turn the paper easier to read